# Optical Enhancement of Diffraction Efficiency of Texas Instruments Phase Light Modulator for Beam Steering in Near Infrared

**DOI:** 10.3390/mi13091393

**Published:** 2022-08-26

**Authors:** Jiafan Guan, Zhipeng Dong, Xianyue Deng, Yuzuru Takashima

**Affiliations:** James C. Wyant College of Optical Science, University of Arizona, 1630 E. University Blvd., Tucson, AZ 85719, USA

**Keywords:** LiDAR, phase light modulator (PLM), MEMS, computer-generated hologram (CGH), beam steering, Talbot image

## Abstract

Phase light modulator (PLM) by MEMS mirror array operating in a piston-mode motion enables a high-speed diffractive beam steering in a random-access and flexible manner that makes a lidar system more intelligent and adaptive. Diffraction efficiency is determined by the range of the piston motion of the MEMS array; consequently, a larger range of the piston motion is required for beam steering in infrared, such as for lidar. We demonstrated how the range of the piston motion is optically enhanced by a factor of two with a light-recycling optics based on Talbot self-imaging. The proposed optical architecture extends the usable range of the wavelength so that a MEMS-PLM designed for visible wavelength is applicable for a high-efficiency beam steering at an infrared wavelength of 1550 nm with an improved diffraction efficiency of 30%.

## 1. Introduction

Laser beam steering (LBS) by arrayed microelectromechanical systems (MEMS) spatial light modulators (SLMs) is an attractive solution for solid-state lidar transmitters and receivers. Diffractive LBS by MEMS-SLMs with computer-generated holograms (CGHs) eliminates bulky mechanical motions commonly found in mechanical scanning modalities such as gimbal mirrors, polygon mirrors, and rotating prism pairs. Recently, a fast MEMS-based phase light modulator (MEMS-PLM) has been introduced [1,2,3]. The MEMS-PLM leverages the semiconductor fabrication process of a sibling, a digital micromirror device (DMD); therefore, it inherits scalability to a large device area, fast operation speed, wavelength polarization diversity, and long-term reliability [4]. In the diffractive LBS, the diffraction efficiency depends on the maximum phase modulation depth of CGHs displayed on the MEMS-PLM. For example, a staircase approximated phase profile with 2π [rad] modulation achieves high diffraction efficiency (DE) close to 100%, in theory [5]. Typically, lidar applications employ a near infrared light source, such as at 905 nm and 1550 nm. Phase SLMs at the NIR wavelength range, liquid crystal (LC) SLMs, achieve high DE; however, they suffer from relatively slow speed in beam steering. The MEMS-SLMs operate even faster, though current generations of the MEMS-PLMs designed for rapidly emerging visible wavelength applications, e.g., holographic projection for head-up display, suffer from a relatively low diffraction efficiency (DE) in infrared. For example, at a wavelength of 1550 nm, the maximum phase modulation depth of a MEMS-PLM is decreased to 0.81π [rad] when we employ a MEMS-PLM designed for 633 nm. Consequently, the diffraction efficiency decreases to about ¼ of its maximum value.

To mitigate the insufficient phase modulation depth of the MEMS-PLM, while fully taking advantage of its fast operation, we propose an optical architecture to enhance the phase modulation depth of the visible and reflective MEMS-PLM by Talbot self-imaging and light-recycling optics. The proposed optical architecture doubles the phase modulation depth which increases diffraction efficiency at 1550 nm by factor of 2. The enhanced diffraction efficiency enables the use of the visible MEMS-SLM for LBS at near infrared with enhanced diffraction efficiency. 

In this paper, we overview an optical architecture of the phase-enhancement technique in Section 2. Section 2 also addresses experimental verification of the theory by visualizing the enhanced phase profile of MEMS-PLM by an interferometer. In Section 3, we address analysis of a critical wavelength, $\lambda_{c}$, above which the enhancement of diffraction efficiency by light recycling overcomes the losses induced by the light-recycling optics. Section 3 also discusses scalability and limitation of the proposed optical enhancement of DE for lidar applications.

## 2. Double-Phase Modulation by Reflective MEMS-SLMs

Figure 1 schematically depicts a principle of an enhancement of the phase modulation depth of a reflective MEMS-PLM. When a periodic phase object is illuminated by a plane wave, the periodic phase pattern is reproduced along the optical axis as Talbot images [6]. With a phase structure having a period of n×p, where n≥0 is an integer and p is pixel period illuminated by a plane wave, Talbot images appears at multiple Talbot distances, $z_{Talbot,\;m}=m\frac{2{\left(pn\right)}^{2}}{\lambda }$, where m is a positive integer, and λ is the wavelength [6]. At $z_{Talbot}$, suppose we place a second reflective MEMS-PLM displaying an identical phase profile to the first one. The phase modulation depth is doubled provided that the Talbot images and the second MEMS-PLM is aligned, or pixel-matched. 

Phase CGHs for laser beam steering generally involve multiple periodicities [7]. The Talbot images of each of the CGHs are set to be appeared at the common distance. To enhance the phase modulation depth over multiple periodicities of CGHs, we set $z_{Talbot}$ so that (*n*, *m*) satisfies the single Talbot imaging condition. In this manner, the phase enhancement is assured for multiple periodicities of CGHs while using a single distance between the first- and second-phase SLMs. The dual interactions of waves with two MEMS-SLMs doubles the phase modulation depth without employing reimaging optics. 

Based on this principle, in lieu of using two MEMS-SLMs, the light-recycling optics depicted in Figure 1b equivalently enhance the phase modulation depth.

A vertically polarized (VP) and collimated light is deflected by a polarized beam splitter (PBS), followed by passing through a quarter-wave plate (QWP) that converts linear polarization (LP) to right circular polarization (RCP). The SLM reflects and modulates phase of light while changing the polarization from RCP to left circular polarization (LCP). The second interaction with QWP changes the polarization of light from LCP to horizontally polarized (HP) light. The mirror M1 is placed at half of the Talbot distance, or $z_{Talbot}/2$. The reflected light by M1 is an HP light; therefore, it goes through the PBS and is converted to RCP by the third interaction with QWP. Finally, the light is doubly modulated by the PLM with the same phase modulation profile and reflected to the direction along the incident laser beam via QWP and PBS. After the laser beam is doubly modulated, diffraction efficiency is increased by the enhanced phase modulation depth. The incoming and outgoing beam has the same polarization. The beam can be separated, for example, by the mirror placed in the 4f 1:1 collimating optics, followed by a collimating optics for beam steering as discussed in a later section.

### 2.1. Visualization of Enhancement of Phase Modulation 

We experimentally visualized phase modulation of the MEMS-PLM, Texas Instruments Phase Light Modulator (TI-PLM) [2]. The TI-PLM is a MEMS-based spatial light modulator. An array of 960 × 540 micromirrors with a 10.8 μm pixel period is electro-statically actuated in piston motion to modulate the phase up to 2π [rad] at 633 nm. Figure 2 shows the phase modulation profile of a binary CGH displayed on a MEMS-PLM for: (a) no phase shift (all-flat state); (b) binary π phase shift with single modulation; (c) π/2 phase shift with dual-phase modulation; (d) π phase shift with dual-phase modulation. Phase modulation scheme, single or double, is selected by rotating the QWP in Figure 1b. Phase after modulation is visualized by a Linnik interferometer [8]. With a single-phase modulation, the fringe shifts by half of the fringe period for the π phase shift. The amount of fringe shift in Figure 2c (half of fringe period) indicates that π phase shift is observed from π/2 phase in the dual-phase modulation setup. Finally, a phase modulation depth of 2π is observed from the π phase, doubly modulated by using the light-recycling optics (Figure 2d).

### 2.2. Diffraction Efficiency in Dual-Phase Modulation

At the wavelength of 532 and 1550 nm, diffraction efficiencies of binary and staircase approximated sawthooth (blazed) phase grating in single and dual modulation schemes is evaluated (Figure 3 and Figure 4). At the period of grating, we evaluated 4-pixel and 10-pixel gratings. Diffraction efficiency (DE) is defined as a ratio of the output power to the input power to and from the light-recycling optics. 

To minimize loss by light-recycling optics, the PBS, QWP, and mirror are AR-coated at each of the wavelengths. As a laser source, a 1550 nm CW laser (LDC-3722B, ILX Lightwave, Bozeman, USA) and a 532 nm laser (DJ532-10, Thorlabs, Newton USA) were used. A 1550 nm laser-power meter (8163A, Hewlett-Packard, Santa Clara, USA) and a 532 nm laser-power meter (1918-R, Newport, Irvine, USA) were used for 1550 nm and 532 nm, respectively. Since input and output beams are counter-propagating and colinear, a beam splitter with a measured and known splitting ratio was placed in front of the PBS to separate the output beam from the input beam. The DE of the light-recycling system is calculated while taking into account the loss of the beam splitter. 

At the wavelength of 532 nm, single modulation exhibits a higher diffraction efficiency (DE) compared to the dual modulation for both binary- and staircase-approximated blazed CGHs. The result indicates that the dual modulation architecture exhibits more loss, as expected. In contrast, at a wavelength of 1550 nm, the dual modulation architecture outperforms the single modulation, for which the phase modulation depth is limited up to 0.8π [rad] at 1550 nm. Since the number of optical surfaces for dual and single modulation are 11 and 5 surfaces, respectively, the dual modulation architecture exhibits more loss. However, double-phase modulation depth (1.6π [rad] at 1550 nm) yields a higher DE by overcoming the loss of optics.

### 2.3. Diffraction Efficiency as a Function of the Grating Period

Figure 5 shows the diffraction efficiency of the binary- and staircase-approximated blazed CGH having a grating period of 4, 5, 6, 7, and 8 pixels at a wavelength of 1550 nm. The roundtrip optical path length between the mirror and the PLM is adjusted to $z_{Talbot}$ = 130.035 mm so that that the system enhances the DE of the CGH with 4-, 6-, and 8-pixel periods. We observed the enhancement in the DE of the CGH at 4-, 6-, and 8-pixel periods. With 5- and 7-pixel periods, enhancement was still observed, but it is not comparable to the CGHs that satisfy the Talbot imaging condition.

As further evidence to show that the dual modulation optical architecture employs Talbot imaging, the system throughput was measured as a function of spacing between the mirror and the MEMS-PLM (Figure 6). For binary phase gratings with 2 and 4 pixels/period, the measurement was conducted by varying the spacing followed by adjusting the tilt angle of the mirror, M1 (Figure 1), so that pixel matching was preserved. System throughput is evaluated by capturing diffraction pattern with an infrared camera (model # C2741-03 Hamamatsu) followed by integrating pixel intensity around the 1st-order diffraction spot. 

The result in Figure 6 shows that peak throughput occurs at half integer of Talbot distance both for 2 pixels/period and 4 pixels/period gratings. The average peak distances are 0.877 (Figure 6a) and 3.5 (Figure 6b) mm for 2 and 4 pixels/period, respectively. The Talbot distance for 2 pixels/period and 4 pixels/period grating are 1.753 and 7.015 mm, which are approximately twice the value in Figure 6. Since the tilt angle of the mirror is adjusted to maximize the system throughput at each of the increments of mirror–MEMS spacing, pixel matching to the conjugate phase reconstructions occurs at half of the adjacent Talbot distances where a laterally shifted Talbot image comes up.

The periodical variation of system throughput indicates that pixel matching takes a role in the mechanism of enhancement of diffraction efficiency. The effect of pixel matching on system throughput is plotted in Figure 7. First, the tilt angle of the mirror was adjusted to maximize the system throughput (pixel-matched), then the tilt angle of the M1 mirror was perturbed by Δθ. At the TI-PLM plane, pixel mismatch between the Talbot image and the physical pixel of the MEMS-PLM is represented by 2ΔθL, where *L* is an optical path length between the mirror and the MEMS-PLM. The average interval of the peaks of system throughput is 0.0217° and 0.0325° for 2- and 4-pixel periods, respectively. With *L* = 130.06 mm, as we have used, the pixel mismatch at the MEMS-PLM plane is 43.4 and 68.6 um, respectively. With the pixel pitch of the MEMS-PLM (10.8 μm), 4- and 6-pixel periods are 43.2, and 64.8 μm, which agrees with the experimentally induced pixel mismatch.

## 3. Discussion

Experimental results in Figure 3, Figure 4 and Figure 5 show that the DE is approximately doubled at 1550 nm by employing the dual-phase modulation. Additionally, Figure 2, Figure 6 and Figure 7 show that Talbot self-imaging takes a role in the enhancement of diffraction efficiency. 

At a shorter wavelength of 532 nm, single modulation yields a higher DE since the MEMS-PLM has sufficient phase modulation depth with a low loss of optics; in particular, the anti-reflection coating of the cover glass of the MEMS-PLM is optimized for a visible wavelength [9]. The question arises of what the wavelength above which the dual modulation optical architecture outperforms the system throughput of the single modulation is. 

It is reported that the diffraction efficiency scales with reflectivity of the MEMS-PLM for all-flat state [1,2,3,7]. The all-flat and 0th-order reflectivity was measured in free space at a wavelength of 532 nm without employing the light-recycling optics. Separately, optical loss of the light-recycling optics is measured by replacing the MEMS-PLM with a high-reflectivity mirror. The result is tabulated in Table 1. 

The 0th-order reflectivity of all-flat MEMS-PLM involves two competing effects: overall surface flatness of micromirror array and transmission of the cover glass [9]. As wavelength increases, surface flatness represented by optical path difference (OPD) decreases. Consequently, the 0th-order reflectivity increases and is approximated by a Strehl intensity ratio (SIR), as follows:(1)$$SIR=1-{\left(\frac{2\pi }{\lambda }\right)}^{2}\sigma^{2}$$
where $\sigma^{2}$ is a variance of wavefront aberration induced by surface flatness of the MEMS-PLM. At a longer wavelength, the transmission of the cover glass material (Corning Eagle XG) $T_{C}$ substantially decreases, since the anti-reflection coating is designed for the visible wavelength, $T_{C}$ > 0.97 at 532 nm. Assuming that the 0th-order reflectivity is mainly due to the wavefront aberration induced by the surface flatness of the entire mirror array, the variance of wavefront error is estimated $\sigma^{2}=0.00287$ [mm^2^]. We define a transmission spectrum function, T(λ), as follows:(2)$$T_{i}\left(\lambda \right)=T_{O,i}\left(\lambda \right)\;{\left(SIR\left(\lambda \right)T_{C}\left(\lambda \right)\;\right)}^{i}$$
where $T_{O,1}\left(\lambda \right)$ and $T_{O,2}\left(\lambda \right)$ are optical transmissions for single and dual modulation schemes, respectively and $T_{C}\left(\lambda \right)$ is transmission of the cover glass. For the single modulation scheme, the input wave interacts with the MEMS-PLM once (*i* = 1) and twice for dual modulation (*I* = 2).

The system diffraction efficiency of single and dual modulation is ($DE_{System,i}\left(\lambda \right)$), estimated by multiplying $T_{i=1,2}\left(\lambda \right)$ with the diffraction efficiency of the sawtooth phase gratings and is given by
(3)$$DE_{System,i}\left(\lambda \right)=T_{i}\left(\lambda \right)DE_{Sawtooth,i}\left(\lambda \right)$$
where
(4)$$DE_{Sawtooth,i}\;(\lambda )=1\;for\;\left(\lambda \leq i\;\lambda_{d}\right),\mathrm{or}\;Sinc^{2}\left(\frac{i\lambda_{d}}{\lambda }-1\right)for\;\left(\lambda >i\;\lambda_{d}\right)$$

In Equations (3) and (4), $\lambda_{d}$ is the wavelength for which MEMS-PLM provides 2π-phase modulation. Figure 8 plots $T_{i}\left(\lambda \right)$ along with $T_{O,i=1,2}\left(\lambda \right)$, SIR(λ), $T_{C}\left(\lambda \right)$, and SIR(λ). Figure 9 plots $DE_{System,i}\left(\lambda \right)$ and $DE_{Sawtooth,i}\left(\lambda \right)$, along with the experimental data plotted in Figure 4.

Figure 8 shows that the overall transmission in dual-phase modulation is lower than that of the single modulation ($T_{2}\left(\lambda \right)<T_{1}\left(\lambda \right)$), due to the higher optical loss and double SIR loss. However, at a wavelength of λ>0.95 μm, the system efficiency of the dual-phase modulation scheme outperforms the system efficiency of the single-phase modulation, since as wavelength increases, the $DE_{Sawtooth,1\;}\left(\lambda \right)$ starts decreasing at 633 nm, while $DE_{Sawtooth,2\;}\left(\lambda \right)=1$ until λ reached 1266 nm. 

Based on Figure 8, we can conclude that the transmission of the single modulation and dual modulation systems is highly related to the transmission of the cover glass of the MEMS-PLM. Figure 9 indicates that the current system diffraction efficiency is restricted by the system loss and surface flatness. By comparing the system and sawtooth diffraction efficiency curves, the system loss plays the predominate role of decreasing the diffraction efficiency. As a result, we expect that increasing the transitivity of the cover glass by appropriate AR coating will substantially increase the system throughput. For example, the transmission of the cover glass of the MEMS-PLM designed for visible wavelength is 70% at λ=1.55 μm. Since the light in a dual modulation setup interacts with the cover glass twice, the loss by the cover glass is about 49% of transmission for dual modulation. If we replace the cover glass with AR coated at λ=1.55 μm, then the diffraction efficiency of the dual modulation can be improved by another factor of 2, or 60%. 

The analysis of the system throughput shows that the dual modulation scheme exhibits a higher system throughput at a wavelength of λ=1.55 μm used for eye-safe automotive lidar [10]. In an actual lidar application, sensitivity of the system throughput to environmental variation such as pixel matching and distance between the MEMS-PLM and mirror is critical. From Figure 6, the tolerance of mirror to MEMS-PLM is roughly 50 um to keep the system throughput higher than 90% of its peak value. Suppose the variation of temperature is 100 degrees C, and with a CTE (coefficient of thermal expansion) of Invar $1.2\times 10^{-6}/K$, the support structure of 50 mm in length deforms 6 um, for which decrease in the DE is negligible. 

The input and output beams are colinear, with the same polarization, but they propagate in opposite directions. To implement the dual modulation in an actual beam steering, separation of the output beam from the input beam is required. The beam is separated by adding a 4f relay with a mirror at its Fourier plane, shown in Figure 10. Input beam is focused at a 45-degree mirror with a hole and re-collimated. The phase modulated beam is picked up by the 45-degree mirror and is redirected for beam steering [11]. 

## 4. Conclusions

MEMS phase light modulators (MEMS-PLMs) operating at a visible wavelength suffer from a lower diffraction efficiency when they are used for infrared applications such as beam steering for infrared lidar. The newly proposed Talbot image-based pixel matching doubles the phase modulation depth by forming a phase image generated by the MEMS-PLM on top of itself with a light-recycling optics. The system diffraction efficiency is doubled at a wavelength of λ=1.55 μm, compared to the conventional diffractive-beam-steering architecture.

With a cover glass of the MEMS-PLM coated for visible wavelength, the proposed approach enhances diffraction efficiency at a wavelength longer than λ=0.95 μm. At λ=1.55 μm, diffraction efficiency of 30% is experimentally confirmed with CGHs having a periodicity of 4, 6, 8, and 10 pixels that enables beam steering in a quasi-continuous manner. The system is two times more efficient compared to the use of a MEMS-PLM in a conventional manner. The diffraction efficiency can be further increased by a factor of two with an AR-coated cover glass at λ=1.55 μm.

Talbot self-imaging-based enhancement of diffraction efficiency extends the applicability of visible MEMS-PLMs to near infrared with minimum additions and modifications: polarizing beam splitter, recycling mirror, quarter-wave plate, and preferably the application an AR coating of the cover glass at the wavelength of interest. Tolerance analysis shows the critical dimension dual modulation setup, spacing between mirror and the MEMS-PLM satisfies the required tolerance to keep a high diffraction efficiency, in particular for variation of temperature range of 100 °C.

The proposed optical architecture is scalable regardless of the design wavelength of the MEMS-PLM. For example, next-generation MEMS-PLMs designed for a wavelength of 1.55 um can be used for applications employing mid-infrared, i.e., beam steering for free-space optical communication and wavefront correction for astronomy with minimum modifications to the optical system.

## Figures and Tables

**Figure 1 micromachines-13-01393-f001:**
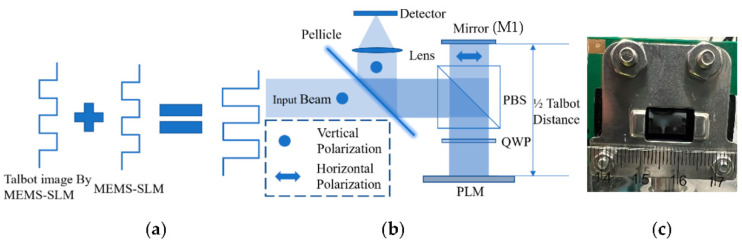
(**a**) Schematic diagrams for pixel matching condition; (**b**) Optical enhancement architecture; (**c**) photograph of the 0.47-inch Texas Instruments Phase Light Modulator.

**Figure 2 micromachines-13-01393-f002:**
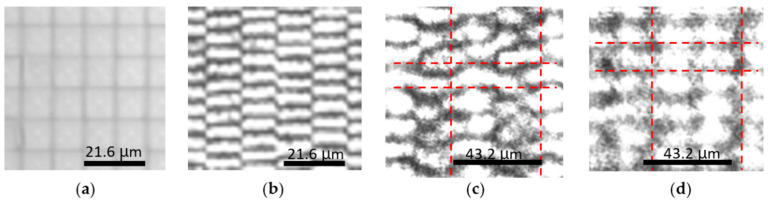
(**a**) Optical microscope image of PLM mirror array; (**b**) Linnik interferometric image of the PLM mirror array, with binary π/2 phase modulated CGH with 2 pixels/period; (**c**) doubly modulated binary π /2 CGH with 4 pixels/period. The half fringe shift within a single period shows phase is modulated by π; (**d**) the doubly modulated π-phase-shifted pattern shows a continuous fringe over pixels that shows that a 2π phase modulation is achieved. For (**a**–**c**), the PLM is illuminated by a collimated 532 nm beam.

**Figure 3 micromachines-13-01393-f003:**
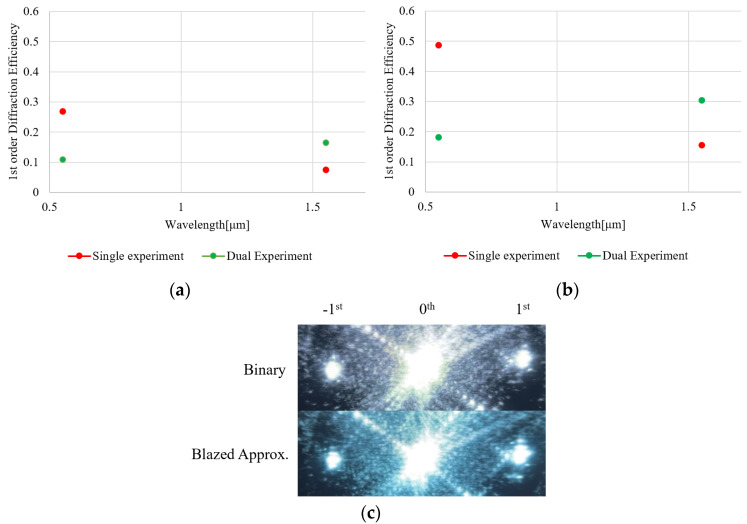
A +1st order diffraction efficiency for 4 pixels/period with (**a**) binary- and (**b**) staircase-approximated blazed grating profiles; (**c**) direct image for grating period of 4 dual modulation for binary and staircase, approximated by IR camera (C274103, Hamamatsu, Hamamatsu, Japan).

**Figure 4 micromachines-13-01393-f004:**
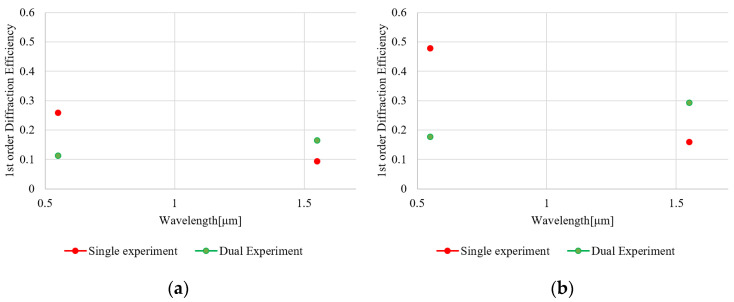
A +1-order diffraction efficiency of CGH for 10 pixels/period with (**a**) binary- and (**b**) staircase-approximated blazed grating profiles. At wavelength of 532 nm, conventional single modulation has higher DE. At 1550 nm, dual-phase modulation increased the DE by a factor of 2.

**Figure 5 micromachines-13-01393-f005:**
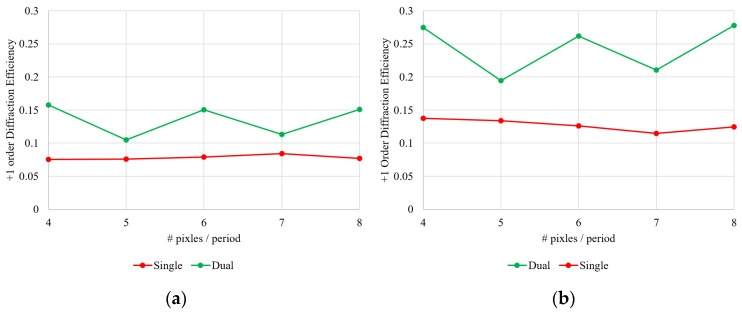
A +1st order diffraction efficiency as a function of the number of pixels/periods between the (**a**) binary- and (**b**) staircase-approximated blazed grating. The spacing between the mirror (M1) and the MEMS-PLM is set to the Talbot distance, which supports grating period of 4, 6, and 8 pixels.

**Figure 6 micromachines-13-01393-f006:**
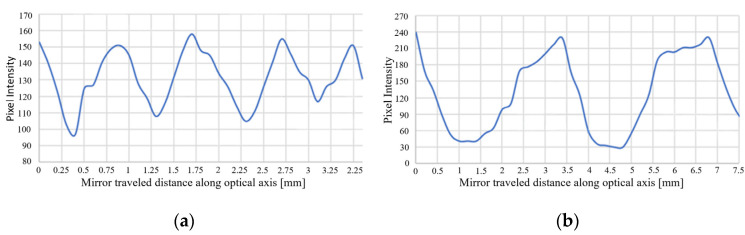
System pixel intensity as a function of spacing between the mirror and the PLM for (**a**) 2 pixels/period and (**b**) 4 pixels/period. (The *x*-axis is starting from an arbitrary position.) System throughput exhibits periodic oscillation at half of the adjacent Talbot distances.

**Figure 7 micromachines-13-01393-f007:**
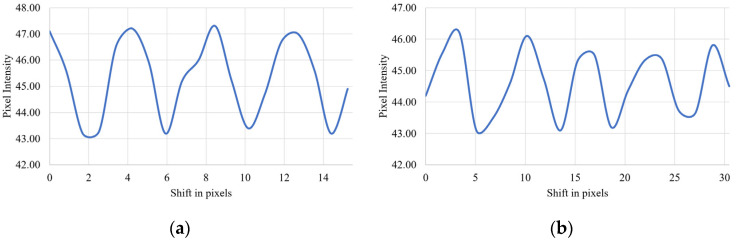
System throughput represented by integrated pixel intensity as a function of lateral shift of the Talbot image by tilting mirror M1 for grating period of (**a**) 4 pixels and (**b**) 6 pixels. The signal exhibits periodic oscillation with periodicity of 4 and 6 pixels.

**Figure 8 micromachines-13-01393-f008:**
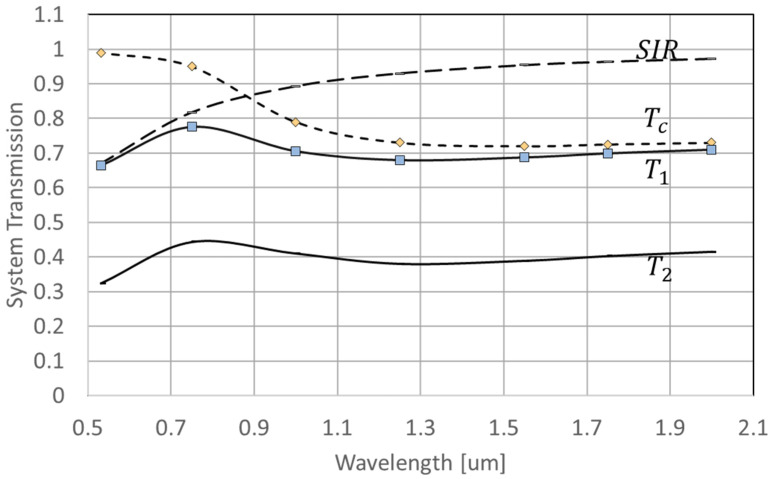
System transmission of single modulation ($T_{1}$) and dual modulation ($T_{2}$ ), plotted as a function of wavelength. The system transmission is a product of cover glass transmission ($T_{C}$ ) and Strehl intensity ratio (SIR ).

**Figure 9 micromachines-13-01393-f009:**
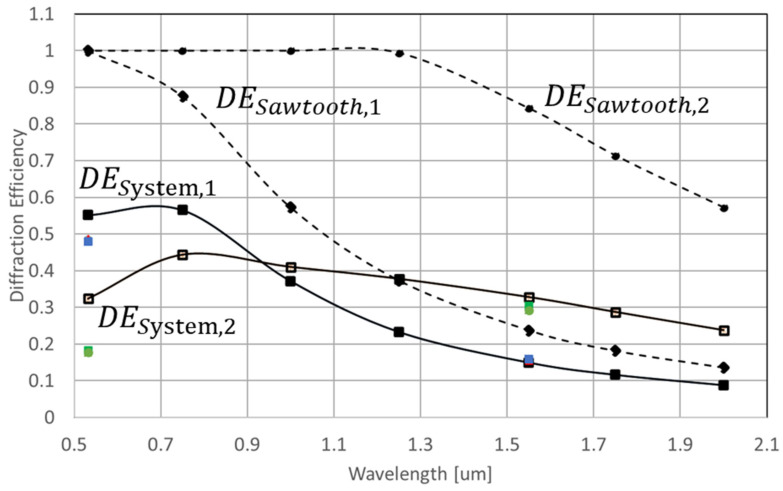
System diffraction efficiency of single modulation ($DE_{System,i=1}$) and dual modulation ($DE_{System,i=2}$ ) is plotted as a function of wavelength and is a product of the system transmission ($T_{i=1}$ and $T_{i=2}$ in Figure 7) and theoretical diffraction efficiency of single- ($DE_{Sawtooth,1}$) and dual-phase modulation (($DE_{Sawtooth,2}$). The green and blue square dots show measured diffraction efficiency of dual and single modulation, respectively.

**Figure 10 micromachines-13-01393-f010:**
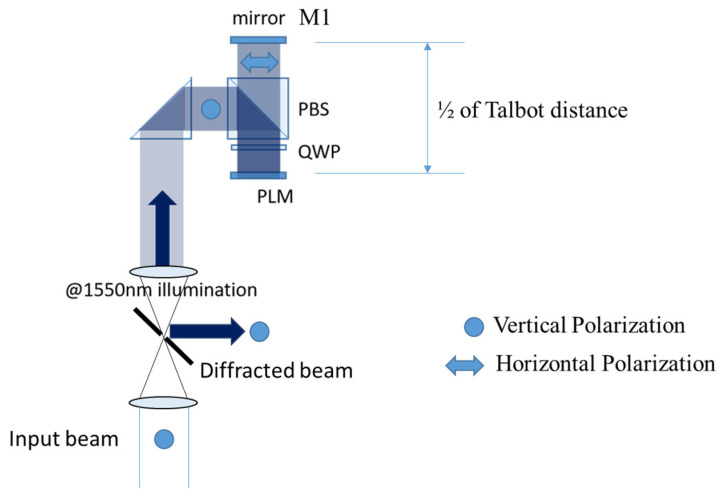
Schematic of the complete dual modulation architecture. Input beam and output beams are angularly separated by a 4f imaging optics and a mirror with a hole placed at the Fourier plane of the 4f relay. The diffracted beam from the dual modulation setup is picked up and redirected for beam steering.

**Table 1 micromachines-13-01393-t001:** Modulation-, binary-, and staircase-approximated blazed grating of MEMS-PLM and optical loss.

Modulation	Single		Dual	
Wavelength	0.532 μm	1.55 μm	0.532 μm	1.55 μm
Binary	0.67	NA	N/A	N/A
Staircase approximated blazed	0.85	0.83	0.73	0.83

## Data Availability

Data is available upon request from the corresponding author.

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
