# Peer review of "Optical Enhancement of Diffraction Efficiency of Texas Instruments Phase Light Modulator for Beam Steering in Near Infrared"

_micromachines, 2022, doi:10.3390/mi13091393_

Round 1

Reviewer 1 Report

The paper is about the enhancement of DE of MEMS phase modulator using light recycling optics. The idea of using Talbot image matching is well proven by experiment. Paper is logically well organized.

But, during reading this paper, I have some questions and comments below. If these are reasonable questions and comments, then I suggest that authors should consider modifing the manusrcipt.

Questions and comments

- I think the baseline is that there is no proper MEMS phase modulator for IR wavelength. If so, authors should mention about these questions: Is it difficult to make it? Why is it so difficult to make it?

- Authors mentioned LiDAR application, so DE enhancement effect for various steering angles should be included.

- For Fig. 8 and 9, 2 data points are not enough to confirm the whole tendency(0.5~2.0 um range) extracted by theory. Can you do more experiments wth other wavelengths in between?

Typos:

- Line98: Texan -> Texas?

- Line265: Eqns -> Eqs?

             2p -> 2pi?

Author Response

Hello, 

Thanks!

Reviewer 2 Report

By employing self-imaging technique, the modulation depth of a MEMS mirror array-based phase SLM of visible light is enhanced at near infrared (1550nm) light, therefore the diffraction efficiency is boosted as well. The main loss comes from non-optimized AR coating for 1550nm and more optical surface contacts. For unmatched z_talbot, there is also an efficiency drop. The idea looks interesting to me, and I recommend this manuscript for publishing after some revisions. Some specific questions are listed below.

Author Response

Hello, 

Thanks

Round 2

Reviewer 1 Report

All the comments and questions from me are mentioned.

I expected more detail description about the difficulties of making IR MEMS phase modulator and the experimental results of steering, but authors just added simple sentenses.

For Fig 8 and 9, number of data points properly added.

Overall, the revised manuscript is improved.